



# Non-target and suspect characterisation of organic contaminants in Arctic air, Part II:
# Application of a new tool for identification and prioritisation of chemicals of emerging Arctic concern in air

Laura Röhler[1,2], Martin Schlabach[2], Peter Haglund[3], Knut Breivik[2,4], Roland Kallenborn[1] and Pernilla Bohlin-Nizzetto[2]

[1]Faculty of Chemistry, Biotechnology and Food Sciences (KBM), Norwegian University of Life Sciences, Ås, Norway
[2]Department of Environmental Chemistry, NILU – Norwegian Institute for Air Research, Kjeller, Norway
[3]Department of Chemistry, University of Umeå, Umeå, Sweden
[4]Department of Chemistry, University of Oslo, Oslo, Norway

*Correspondence to*:  Laura Röhler (laura.rohler@nmbu.no)

**Abstract.** The Norwegian Arctic possess a unique environment for the detection of new potential chemicals of emerging Arctic concern (CEACs) due to remoteness, sparsely populated and the low number of local contamination sources. Hence, a

contaminant present in Arctic air is still considered a priority indication for its environmental stability and environmental mobility. Today, legacy persistent organic pollutants (POPs) and related conventional environmental pollutants are already well-studied since their identification as Arctic pollutants in the 1980s. Many of them are implemented and reported in various national and international monitoring activities including the Arctic Monitoring and Assessment Program (AMAP). These standard monitoring schemes, however, are based on compound specific quantitative analytical methods. Under such

conditions, the possibility for identification of hitherto unidentified contaminants is limited and randomly, at the best. Today, new and advanced technological developments allow a broader, unspecific analytical approach as either targeted multi-component analysis or suspect and non-target screening strategies. In order to facilitate such a wide range of compounds, a wide-scope sample clean-up method for high-volume air samples, based on a combination of adsorbents was applied, followed by comprehensive two-dimensional gas chromatography separation and low-resolution time-of-flight mass spectrometric

detection (GC×GC-LRMS). During the here reported study, simultaneous non-target and suspect screening were applied. The detection of over 700 compounds of interest in the particle phase and over 1200 compounds in the gaseous phase is reported. Of those, 62 compounds were confirmed with reference standards and 90 compounds with a probable structure (based upon mass spectrometric interpretation and library spectrum comparison). These included compounds already detected in Arctic matrices and compounds not detected previously (see also Figure 1). In addition, 241 compounds were assigned tentative

structure or compound class. Hitherto unknown halogenated compounds, which are not listed in the used mass spectral libraries, were also detected and partly identified.





## 1 Introduction

A high number of organic chemicals is used today in large quantities. By 2019, the Chemical Abstracts Service (CAS) registry[SM], contained more than 156 million unique inorganic and organic chemicals. This is 50% more than in 2015, when CAS was celebrating 100 million registered compounds (Wang, 2015). For the effective regional control of chemicals in

commerce, the REACH register was introduced in the EU-region (Regulation (EC) No 1907/ 2006 of the European Parliament and of the Council concerning the registration, evaluation, authorisation and restriction of chemicals) managed by the European Chemicals Agency (European Parliament, 2018). REACH has only classified about 2000 substances (about 40 % of chemicals registered with a production volume above 100 tonnes per year) into classes of high concern. Such chemicals were identified as carcinogenic, mutagenic, toxic for reproduction (CMRs), persistent, bioaccumulative and toxic (PBT), very persistent and

very bioaccumulative (vPvB) and/ or endocrine disruptors (EDCs) (data status May 2018, (ECHA, 2019b)). The assessment of chemicals with lower production volumes will follow. A considerable amount of organic chemicals is released into the environment by various pathways including insufficient waste management, direct application (e.g. agriculture, structure treatment), unintended by-products from largescale production lines and primary emission/ releases from products and applications. Some of these organic chemicals are persistent and can migrate over long distances, ultimately reaching remote

areas, such as the Arctic (Lebedev et al., 2018; Macdonald et al., 2000; Macdonald et al., 2005; Genualdi et al., 2011; Barrie et al., 1992). An important pathway for long-range transport of persistent organic chemicals is via the atmosphere (Xiao et al., 2012; Genualdi et al., 2011; Hung et al., 2010; MacLeod et al., 2005; Koziol and Pudykiewicz, 2001; Barrie et al., 1992). Environmental persistence and long-range atmospheric transport potential (LRATP) (Zhang et al., 2010; Czub et al., 2008) are two hazard criteria which characterise persistent organic pollutants (POPs). POPs are today considered as priority pollutants

and their use and production is regulated through international agreements, such as the Stockholm Convention on POPs and the Aarhus protocol on POPs under the Convention on Long-range Transboundary Air pollution (CLRTAP) (UNEP, 2009b; UNECE, 1998). In order to evaluate the effectiveness of these agreements aiming at reducing human and environmental exposure to POPs (Fiedler et al., 2019), air monitoring strategies for legacy POPs have been established on national, regional and global levels. Examples are the European Monitoring and Evaluation Programme (EMEP, 2019) for the Aarhus protocol

on POPs (UNECE, 1998), the Global Monitoring Plan (GMP) for the Stockholm Convention (UNEP, 2009a) and the Arctic Monitoring and Assessment Programme AMAP (2019) for the Arctic. Within these, air monitoring of POPs in remote areas including the Polar Regions are used to study the long-range atmospheric transport of POPs to remote areas and such knowledge is considered vital for the understanding of the environmental behaviour of POPs and further international POP regulation. Recently, chemicals of emerging Arctic concern (CEACs) (AMAP, 2017) including new flame retardants,

plasticizers, per- and polyfluoroalkyl substances (PFAS), pharmaceuticals and personal care products (PPCPs), current use pesticides (CUPs) and more, have received increased attention within AMAP. Selected CEACs have already been included in some of the national and regional air monitoring programmes in the Arctic (AMAP, 2009, 2017). Measurements of CEACs in the Arctic provide authorities with crucial knowledge supporting adequate policy measures and, if necessary, national or





international regulations to come into place. In addition, it is important to identify new CEACs in the Arctic at an early stage. While this is often accomplished using biotic matrices there is also a need for measurements in abiotic matrices like air as not all CEACs bioaccumulate but still are persistent and transported over long distances. Non-target and suspect screening (NTS and SUS) approaches represent promising strategies for identification of so far unidentified CEACs. However, standard
sampling and analytical methods used for targeted monitoring of POPs in air are not necessarily suitable for non-target analyses and methodological challenges remain to be solved. For example, some CEACs may have similar properties to legacy POPs while others might be less stable under certain conditions, such as being acid labile (e.g. some flame retardants, cyclic methyl siloxanes as well as some legacy POPs like dieldrin and related compounds) (Röhler et al., 2020). It is, therefore, important to develop non-destructive sample clean-up procedures, e.g. without sulfuric acid, to preserve an expanded range of compounds
for SUS/ NTS strategies in atmospheric samples. As a natural consequence of a wide-scope sample clean-up method, the resulting analytical extracts contain a larger load of interfering background matrix. It is therefore essential to increase the separation power of the instrumental analysis. This could be achieved by high-resolution chromatographic separation and/ or high-resolution mass separation, i.e. high-resolution mass spectrometry (HRMS) methods.

In this study, a new, non-destructive wide-scope sample clean-up procedure and a powerful instrumental analysis method was
applied on high-volume air samples, from an Arctic background monitoring station, aiming at identifying regulated POPs, known CEACs and emerging or new CEACs. The final separation and detection method was comprehensive two-dimensional gas-chromatography (GC×GC), which offers enhanced peak capacity as compared to conventional GC and a better separation of matrix residues from analytes, and low resolution time-of-flight mass spectrometry (LRMS) (Röhler et al., 2020). New potential CEACs were evaluated by comparing them to the PBT classification of the Stockholm Convention (UNEP, 2009b)
with a focus on long-range atmospheric transport potential (LRATP).

## 2. Experimental Section

### 2.1 Air sampling and sample clean-up

Two air samples were collected at the Zeppelin Observatory, on Svalbard (78° 55' N, 11° 53' E, 474 m a.s.l.) in December 2015. Zeppelin is a Norwegian background station providing environmental monitoring data including organic environmental
pollutants to many national authorities and international monitoring programmes; EMEP, AMAP and GMP. The particle phase of the air samples was collected on glass fibre filters (GFFs; 142 mm i.d.; cut-off 10 μm) and the gas phase were collected on polyurethane foam (PUF) plugs (11 cm in diameter, 5 cm in height) using high volume air samplers (average 25 m³ h⁻¹). The sampling time was 4-5 days resulting in sample volumes of 2700 m³ and 3500 m³. Details on the sampling methodology can be found in Kallenborn et al. (2013).
Before extraction, the PUFs from the two air samples were combined in one Soxhlet extractor and spiked with internal standards (ISTDs, details in Table S1, SI). The same was done for GFFs from the two air samples. PUFs and GFFs were Soxhlet extracted separately for 8 h in acetone/ *n*-hexane (1:1 v/v). This resulted in one pooled PUF extract and one pooled

GFF extract. The individual extracts were reduced to 0.5 mL with a Zymark TurboVap and solvent exchanged to isooctane. For clean-up, three-layer liquid chromatography columns were used, with the bottom layer consisting of a mixture of Z-Sep⁺ & DSC-18, the middle layer of Florisil, and the top layer of sodium sulphate. Samples were applied in isooctane and eluted with acetonitrile (ACN)/ 0.5 % citric acid (w/w). Detail about the sample clean-up can be found in the and in the Supplementary

Information (SI) and Röhler et al. (in preparation, (2020)).

## 2.2 GC×GC-LRMS Analysis

The samples were analysed using a LECO Pegasus® 4D, St. Joseph, MI, USA) GC×GC-LRMS system, operating in EI mode. The GC was equipped with a Restek (Bellefonte, PA, USA) Siltek Guard column (4 m, 0.25mm), a SGE (Trajan Scientific and Medical, Ringwood, VIC, Australia) BPX-50 (25 m, 0.25 mm, 0.25 µm) first dimension column and an Agilent J&W

(Folsom, CA, USA) VF-1ms (1.5 m, 0.15 mm, 0.15 µm) second dimension column. Helium (5.0 quality, Nippon gases Norge AS, Oslo, Norway) was used as carrier gas with a constant flow of 1 mL min⁻¹. Three microliter (µL) of each extract was injected into a PTV (programmed temperature vaporiser) inlet, operating in solvent vent mode. For identification of unknown halogenated compounds (see sect. 3.7), the samples were also analysed using a LECO GC-HRT GC×GC-HRMS instrument, operating under the same conditions described above for the GC×GC-LRMS analyses. Details on chromatographic conditions

can be found in the SI.

## 2.3 Quality control

Laboratory blanks, consisting of unexposed PUFs and GFFs, were extracted, cleaned and analysed along the same sample preparation scheme as the exposed samples. The blanks were used for quality assurance, to ensure that identified/ reported compounds have their origin in the collected air sample and do not appear in the blank samples above predefined levels (see

sect. 2.4). This means that compounds need to exceed the area threshold of a factor 100 compared to the area in the sample blanks.

The used ISTDs, which are covering a wide area of the GC×GC chromatogram, were not used for target quantification, but for quality assurance and sample normalization. For example, the early eluting ISTDs (e.g. ¹³C₆-labelled hexachlorobenzene (HCB) or ²H₁₀-labelled phenanthrene) help to identify potential evaporative losses during clean-up and volume reduction, and

the ¹³C₁₂-labelled *p,p'*-dichlorodiphenyltrichloroethane (*p,p'*-DDT) ISTD provides information about possible matrix-effects in the injector and/ or GC-column due to its higher thermal degradation potential. Thus the *p,p'*-dichlorodiphenyldichloroethylene/ *p,p'*-dichlorodiphenyldichloroethane (*p,p'*-DDE/ *p,p'*-DDD) ratio was used for identification of injector losses. A comprehensive recovery test was done by Röhler et al. (2020) to investigate the applicability of this wide-scope sample clean-up method.



## 2.4 Data processing/ Post-acquisition data treatment

For GC×GC-LRMS system control, data analysis and processing, LECOs® ChromaTOF® software (V 4.50.8) was used; including its advanced features, Statistical Compare and Scripts. Several in-house libraries with mass spectra of reference standards, $^{13}C/ ^2H$-labelled ISTDs, National Institute of Standards and Technology (NIST) NIST 2014 mass spectral library,
Scientific Working Group for the Analysis of Seized Drugs (SWGdrug (Oulton, 2019)) mass spectral library, and a customised library with selected spectra from NIST 2014 for suspect screening were used for tentative identification of detected compounds. To create the customised library with selected spectra from NIST14, all mass spectra of compounds from NIST 14, which are listed on relevant suspect lists for the Arctic (Reppas-Chrysovitsinos et al., 2017; Brown and Wania, 2008; Coscollà et al., 2011; Hoferkamp et al., 2010; Howard and Muir, 2010; NORMAN-network, 2019; Vorkamp and Rigét, 2014;
Zhong et al., 2012), were copied to an own library file for more efficient suspect screening. This customised library was useful to detect and flag potential suspects during data processing. More details can be found in (Röhler et al., 2020) and a short description on how the data from suspect lists got aligned with our peak table as well as how the suspect MS libraries were built can be found in the SI.

The identification level classification concept of Schymanski et al. (2015), originally developed for liquid chromatography
(LC)-HRMS data, is defining a common set of rules for harmonised communication of identification confidences of results from different SUS/ NTS studies. Due to the lack of HRMS data in the current study, this level classification concept had to be slightly to account for the limitations of LRMS data (Figure 2), c.f. Röhler et al. (2020). As LRMS analysis does not provide accurate masses, the lowest level of identification confidence, Level 5 (L5), is defined as peaks of interest, which are only characterized by retention time and a mass spectrum, and not by tentative molecular weights. The remaining levels for
identification confidence with LRMS are in line with the original concept of Schymanski et al.: Level 4 (L4), defined by a possible molecular formula, e.g. a plausible molecular formula could be assigned to various compound classes, or halogen cluster detected without match to the used MS libraries. Level 3 (L3), the group of tentative candidates, which are identified as substructure/ class or a certain base structure is possible, e.g. the MS shows fragment patterns of a polycyclic aromatic hydrocarbon (PAH) with a plausible molecular formula but several alternative structures are possible. Level 2 (L2), the group
of probable structures based on good library matches and additional evidence, e.g. the position or grouping on the two-dimensional GC×GC plan. Level 1 (L1) is defined by compounds confirmed by external reference standards. We introduced an additional Level 0 (L0) for compounds confirmed by ISTDs and where target quantification could be performed together with SUS and NTS. Target quantification was however not a primary aim of this study.

During SUS and NTS data processing (Figure 3), the forward match percentage to the mass spectrum (MS) library entry was
used to reduce the number of peaks which require manual inspection. This is a critical step where potential compounds of interest may be lost, since the MS from the NIST14 library are not identical with the MS obtained with the GC×GC-LRMS, probably due to the unit mass resolution of the instrument, generating mass artefacts as shown in Figure 4. Compounds with higher mass defects, e.g. the brominated compounds, had non-acceptable spectra match quality (Figure 4). It is possible that





some compounds of interest were rejected during data processing due to bad match of MS to NIST14 MS library or custom suspect libraries. To minimise such losses of compounds with higher mass defects, visual basic scripts, developed by Hilton et al.(2010), were applied for data processing. These scripts were specifically written for isotope clusters obtained from the used instrument. All compounds flagged by those scripts were checked manually. Furthermore, it was not possible to use

available retention indices for further identification confidence due to the use of a medium polar GC column (BPX-50, 50 % phenyl polysilphenylene-siloxane) as first column for GC×GC separation instead of a non-polar (5 % phenyl) column, for which most of the retention indices are present in databases. In addition, there are limited concepts for the adaption of retention indices for GC×GC, e.g. (Veenaas and Haglund, 2018). This BPX-50 column, as first column for GC×GC separation, was chosen to get a better separation from compounds of interest to interfering background matrix and thus minimise negative

effects on collected mass spectra.

When a compound was flagged in the result list (L1–L5 lists, Figure 3) for manually check after data processing, additional plausibility checks will be performed. These included the selectivity of the sampling and sample clean-up method as well as the complete sample analysis procedure. For instance, a compound should not degrade during sample processing (from sampling to analysis), not evaporate or sorb to the vial, injector or chromatographic column. The GC×GC retention times

should also be reasonable, e.g. volatile compounds cannot elute at the end of the run and non-polar compounds cannot have a short second dimension retention time. Furthermore, the area of a candidate in a sample should exceed the area threshold of factor $\geq 100$ in the corresponding sample blank to be kept in the peak table and not to be sorted out as compound occurring from the blank sample. The higher threshold is necessary since areas are not adjusted for different sample volumes.

## 2.5 Evaluation of long-range atmospheric transport potential

The detection of a substance in air at Zeppelin does not provide conclusive evidence for long-range atmospheric transport. Yet, an organic chemicals potential for LRAT into the Arctic requires that it is sufficiently persistent in air. LRATP can be estimated from theoretical calculations. The key mechanism which is believed to degrade organic chemicals in the atmosphere is reaction with OH-radicals. Because both concentrations of OH-radicals and temperatures are very low during the polar night, the atmospheric half-life due to atmospheric reaction ($t_{1/2}$) is predicted to be very long in comparison to lower latitudes (e.g.

Webster et al. (1998)). For a more realistic evaluation of LRATP, reaction half-lives in air therefore need to be adjusted reflecting the actual sampling conditions. Half-lives were adjusted using an equation from Wania et al (2006) and we refer to the SI for details. To parameterise this equation, the reaction rate in air at 25 °C were retrieved for L0, L1 and L2 compounds from the EPIsuite software (U.S.EPA, 2019) and adjusted using the maximum temperature during sampling (-2.4 °C), an assumed OH-radical concentration of 6E3 mol cm$^{-3}$ and an assumed activation energy for reaction in air of 10000 J mol$^{-1}$.

Estimates of OH radical concentration was based on a model developed by Bahm and Khalil (2004). However, this model does not predict OH-radicals at higher latitudes than 45° N, which crosses central Europe ([OH] at 45° N: 5E4 mol cm$^{-3}$), in December. Our samples were collected at 78° N, and our assumed OH-radical concentration of 6E3 mol cm$^{-3}$ was chosen as an initial conservative estimate, keeping in mind that our analysed air samples include air masses which may have been





transported from lower latitudes. Results from these theoretical calculations are discussed in sect. 3.5.3 and shown in the SI (Table S3 and Excel-SI).

## 3. Results and Discussion

### 3.1 Number of detected and classified compounds in Arctic air

By applying the wide-scope clean-up based on C18 silica and Z-Sep$^+$ combined with Florisil to the air sample extracts from PUFs and GFFs, we were able to expand the chemical domain covered as compared to established target POP analysis methods, which generally are using concentrated sulfuric acid. Our method covers a broad spectrum of polarity, has sufficient matrix removal and is, for the first time, applied on Arctic air samples for the detection and identification of known and new potential CEACs. Previously, this method has been successfully applied to air samples from southern Norway (Röhler et al., 2020).

It was possible to detect and classify over 700 compounds in the particle phase (GFF samples) and over 1200 compounds in gas-phase (PUF samples) as L5 with our classification and sorting method (details on the peak reduction during data-processing for SUS and NTS, Fig. S1 in SI). The higher number of gas phase compounds was expected since particle related compounds, collected on GFFs, may have a lower LRATP compared to gas-phase related compounds, collected on PUFs. Of these L5 compounds, approximately 200 compounds in GFFs and approximately 400 compounds in PUFs could be further classified to

L4, L3 or L2 (Figure 5). As the structures of the remaining L5 compounds remain unknown, these compounds are not discussed any further. In total, 65 compounds (14/51 GFF/PUF) were classified as L4. Many compounds of the L4 class could be identified as unknown halogenated compounds as a halogen pattern was observed, but no match in MS libraries were found (12/29 GFF/PUF). For the remaining L4 compounds, only a possible molecular formula could be assigned. As L3, 241 compounds (95/146 GFF/PUF) could be classified, including two major sub-groups, polycyclic aromatic compounds (PAC)

and phthalates (see Figure 6). The PAC sub-group include many PAHs. Ninety compounds reached L2 (20/70 GFF/PUF) and 41 of the compounds in PUF were PCBs with 2-7 chlorine substituents. By analysing reference standards under identical conditions as the air samples, 56 compounds could be classified as L1 (14/42 GFF/PUF) (Table 1). Furthermore, six compounds could be identified and confirmed with ISTDs to L0 in the PUF sample (only traces in the GFF sample). Of the 56 confirmed L1 compounds, seven were common to GFF and PUF sample. Importantly, a compound not positively confirmed

by this method does not necessarily mean that it does not occur in Arctic air.

As shown in Table 1, 39 of 56 compounds that were classified as L1 are listed in one or more suspect lists (Reppas-Chrysovitsinos et al., 2017; Brown and Wania, 2008; Coscollà et al., 2011; Hoferkamp et al., 2010; Howard and Muir, 2010; NORMAN-network, 2019; Vorkamp and Rigét, 2014; Zhong et al., 2012) or self-built suspect libraries. From L2 compounds, 17 compounds resemble compounds in one or more suspect lists. Since L2 compounds are not confirmed with reference

standards, those compounds might be different isomers than those listed in the SI (Excel-SI) file and thus matches to suspect lists could be different for L2 compounds.





For a better understanding about the importance of our findings at L0, L1 and L2, these compounds were further arranged into four groups: (i) legacy POPs and PAHs, (ii) CEACs defined in the AMAP report (2017), (iii) organic compounds that previously have been detected in Arctic media, and (iv) new potential CEACs not reported in Arctic media to date (October 2019). The new potential CEAC group was split into two subgroups, those with an estimated LRATP and those without. The

default LRATP estimates are based on the EPIsuite software (U.S.EPA, 2019), reflecting standardised environmental conditions ($t_{1/2}$(air) at 25 °C, 12 h days and a hydroxyl radical concentration of 1,6E6 OH cm$^{-3}$) and results compared with the criteria in the Stockholm Convention (UNEP, 2009b) that substances with a $t_{1/2}$(air) exceeding 2 days has a LRATP. A complete table with all compounds identified, including physical-chemical properties from EPIsuite, adjusted half-life in air during sampling (Eq.S1 and Eq. S2, SI), usage and information on previous reports on occurrence in Arctic environments,

toxicity and presence in HPV lists of the EU and US as well as further parameters for PBT classification (REACH and Stockholm conventions) can be found in the SI (Table S2 and Excel-SI).

## 3.2 Legacy POPs and PAHs

The currently used method revealed 59 legacy POPs and PAHs as L0, L1 and L2, specifically hexachlorocyclohexanes (α-HCH and γ-HCH), HCB, pentachlorobenzene (PeCB), DDTs (*o,p'*-DDT, *p,p'*-DDT and *p,p'*-DDD), PCB-153, dieldrin, *trans*-

nonachlor, *cis*-chlordane, PBDE-28 and PBDE-47 and a metabolite of heptachlor (heptachloro *exo* epoxide) (UNEP, 2009b) as L0 or L1. Furthermore, two PAHs, benzo[*ghi*]fluoranthene (L1) and naphthalene (L2) could be identified. Other PAHs were classified as L3 (PAC). Dieldrin and benzo[*ghi*]fluoranthene were common to GFF and PUF and had GFF:PUF ratio according to Peak area of 1:8 for dieldrin and 2:1 for benzo[*ghi*]fluoranthene. It was also possible to classify 41 PCB congeners as L2. The finding of legacy POPs and PAHs, routinely measured at the same monitoring station using target methods, is an indirect

validation of the method and indicates that detection of other compounds with similar physical-chemical properties are trustworthy. From the assumption that a higher concentration of a compound gives a greater peak area, the detected legacy POPs could be correlated with a good match to the average concentrations of monitored legacy POPs at the Zeppelin station (Table 2) (Nizzetto, 2016). Pearson correlation analysis indicates a strong correlation (r = 0.978) that is significant different from zero (p < 0.001). Thus, the screening approach seems to give an indication of the relative concentrations (occurrence) for

semi-volatile organic compounds in Arctic air.

## 3.3 CEACs as defined by AMAP

Eleven of the detected compounds are included as CEACs in the AMAP report (2017) or in Reppas-Chrysovitsinos et al. (2017). One was classified as L0, five were classified as L1 and five were classified as L2. The CEAC, classified as L0 was the flame retardant hexabromobenzene (HBB) that also have been detected in air at Zeppelin Observatory by target analyses

as a part of the Norwegian national air monitoring programme for long-range atmospheric transported contaminants. Classified as L1 were two halogenated natural products (HNPs), 2,4,6-tribromoanisole (TBA) and 2,4-dibromoanisole (2,4-DBA), the pesticide metabolite pentachloroanisole (PCA), the organophosphorus flame retardants (OPFRs) tri(2-chloroethyl) phosphate





(TCEP) and the stimulant caffeine. The five L2 compounds were the BFR pentabromotoluene (PeBT), on isomer of TCEP, two isomers of tris(2-chloroisopropyl) phosphate (TCPP), and an isomer of dibromoanisole (DBA), likely the HNP 2,6-DBA. TBA is routinely measured in air at the Zeppelin observatory as a part of the Norwegian monitoring programme. TBA has also been reported earlier in Arctic air from the Zeppelin station by Vetter et al. (2002). Bidleman et al. detected 2,4-DBA and TBA

at Pallas, Finland (Bidleman et al., 2017a) as well several locations at the Bothnian Bay region (Bidleman et al., 2017b). PCA is a pesticide metabolite, originating from biodegradation of the pentachlorophenol, which is a pesticide and wood preservative (GovCanada, 2019; Su et al., 2008). PCA has previously been found in air at other AMAP sampling sites, like Alert, Canada, but not at Zeppelin, Svalbard (Su et al., 2008; Hung et al., 2010). The stimulant and food additive caffeine, also an intermediate for pharmaceuticals as well as perfumes, fragrances, personal care products and laboratory chemicals (ECHA, 2019h), was

found in effluent and seawater from Longyearbyen (Kallenborn et al., 2018) but to our knowledge not in air samples. TCPP (ECHA, 2019i; Sühring et al., 2016) is one of the main substances which have replaced TCEP in Europe (UK, 2008). TCPP and TCEP were detected in our GFF sample (i.e. particle phase), together with structurally related isomers. OPFRs have previously been detected in Arctic air from the Zeppelin Observatory (Nizzetto, 2018; Salamova et al., 2014).

### 3.4 Organic compounds, previously detected in Arctic media

Besides legacy POPs and PAHs, and CEACs listed by AMAP, it was also possible to identify eight other organic compounds as L1 and classify one compound as L2. These nine compounds have previously been reported in Arctic samples. As L1 we found tetrachloroveratrole, octachlorostyrene (OCS), 1,2,3,4-tetrachlorobenzene, 1,9-benz-10-anthrone, 9-fluorenone, 9,10-anthraquinone and 4H-cyclopenta[*def*]phenanthren-4-one. Only one isomer of tetrachloroveratrole was classified as L2. Tetrachloroveratrole, and its isomer, are both pesticide metabolites (Su et al., 2008; GovCanada, 2019), while the others

were either combustion products or oxidation products of PAHs (Kirchner et al., 2016; Su et al., 2008; Hung et al., 2010; Gubala et al., 1995; Singh et al., 2017; Karavalakis et al., 2010). 4H-Cyclopenta[*def*]phenanthren-4-one was common to GFF and PUF with a GFF:PUF ratio from peak areas of 1:2. Tetrachloroveratrole and OCS have been reported from other Arctic monitoring sites like Alert, Canada, but are not included in the Norwegian monitoring programme at the Zeppelin Observatory on Svalbard (Hung et al., 2016). OCS has also been detected in air samples from the Alps (Kirchner et al.,

2016), 1,2,3,4-Tetrachlorobenzene has been measured in sediments in Arctic Alaska (Gubala et al., 1995), but to our knowledge not in Arctic air before. 1,9-benz-10-anthrone, 9-fluorenone, 9,10-anthraquinone and 4H-cyclopenta[def]phenanthren-4-one have been reported in aerosols, total suspended particles, from the Alert station, Canada (Singh et al., 2017). Besides that, they were detected, among further oxy-/nitro-/ PAHs, in the emissions from a local point source in Longyearbyen, Svalbard (coal fired power plant) (Drotikova et al., 2020). Most of the known Arctic contaminants

were classified as L1 as a result of available standards. Please note, most of PAHs are classified as L3 compounds due to the lack of single reference standards. We assume that several of the known PAHs, previously detected in Arctic media, could be found among the PAHs, classified as PAC in L3 (see section 3.6.).





### 3.5 New potential chemicals of emerging arctic concern

It was possible to classify 73 new potential CEACs with a match to reference standards (L1) or probable structures (L2). These 73 compounds have, to our knowledge, previously never been reported in Arctic media. The complete list can be found in SI (Excel-SI). Almost 40 % of these new potential CEACs have a LRATP according to the Stockholm convention (UNEP, 2009b), $t_{1/2}$(air) exceeding 2 days, using the standard values from EPI suite calculation (see section 3.1.) Although those compounds were not reported in Arctic environment before, local sources cannot be excluded for some of the identified compounds. Especially compounds which might be of biogenic origin, i.e. methoxy-chloro compounds, or compounds with a widespread use, the potential for local sources need to be kept in mind. This study, however, is not designed to prove the potential influence of local sources on the overall contaminant patterns. Especially for compounds that could be HNPs, but for which we could not find any evidence that they have been detected in the Arctic before, further in-depth studies are required.

### 3.5.1 Potential CEACs with LRATP

Out of the total of 73 identified or tentatively identified new potential CEACs, 29 were classified as compounds with LRATP according to the Stockholm convention criteria (UNEP, 2009b), $t_{1/2}$(air) exceeding 2 days, using the standard values from EPI suite calculation. Of these, six compounds were detected in the GFF sample (two as L1 and four as L2) and 23 compounds were detected in the PUF sample (13 as L1 and 10 as L2), see Table 3 and Table 4. Further information about these compounds can also be found in SI (Excel-SI). As the identities of L2 compounds was not fully confirmed, no literature search was performed for previous reports on occurrence in Arctic environments.

In the GFF sample, one of the two L1 compounds was benzenesulfonamide (BSA), an industrial intermediate used for the synthesis of chemicals in commerce like pesticides, photochemical products, pharmaceuticals, sweeteners or dyes (ECHA, 2019g; Naccarato et al., 2014; Herrero et al., 2014). Since BSA occurs in many products, local sources cannot be excluded and further investigations are needed to confirm a potential LRATP or local sources as major contamination source of BSA in the here investigated sample. The other L1 compound identified in the GFF is a potential combustion product, 2-methyl-9,10-anthraquinone, which can have its origin in wood combustion (Czech et al., 2018; Lui et al., 2017; Vicente et al., 2016) or can be formed by atmospheric reactions (Alam et al., 2014). 2-Methyl-9,10-anthraquinone is also an intermediate in the production of coating products, inks and toners, laboratory chemicals and explosives, or is also used for the production of plastic products (ECHA, 2019c). Beside those L1 compounds it was possible to detect one 3,4-dichloropropiophenone related compound, likely a positional isomer, three sulphur related compounds, diphenyl sulfone, dibenzothiophene sulfone and N-(2-cyanoethyl)-N-methyl-benzenesulfonamide and classified these as L2 by MS library matching.

In the PUF sample, the pesticide dichlobenil (2,6-dichlorobenzonitril) was identified, together with an isomer, 2,4-dichlorobenzonitrile (ECHA, 2019d), as L1. No information of commercial application and usage is found for 2,4-dichlorobenzonitrile. Besides dichlobenil, another pesticide, chloroneb (1,4-dichloro-2,5-dimethoxybenzene) (U.S.EPA, 2005) was identified as L1, and two chloroneb and one chlorothalonil related compounds, likely positional isomers of those, was





assigned L2. The nitrification inhibitor, nitrapyrine (2-chloro-6-(trichloromethyl)pyridine), L1, were identified in Arctic samples for the very first time (DOW, 2012; ECHA, 2019e; Woodward et al., 2019). Furthermore, two trichloro-dimethoxybenzenes, two dichloro-methylanisols, and one dibromo-dimethoxybenzene were also assigned L2.

Biogenic origin cannot be excluded for those halogenated methoxybenzenes. Local sources also cannot be excluded for the

closely related 2,4-dichloroanisole and , 2,4,6-trichloroanisole (both L1), potential metabolites of chlorophenol and chlorophenoxy pesticides, but also potential HNPs (Führer and Ballschmiter, 1998; Schenker et al., 2007; Bendig et al., 2013). 2-Naphthalenecarbonitrile, originating most probably from plastic combustion, e.g. ABS (acrylonitrile butadiene styrene) plastic or polyester fabrics (Moltó et al., 2009; Watanabe et al., 2007; Wang et al., 2007; Moltó et al., 2006) or bluing of steel (Stefanye, 1972), was identified as L1, and 1-naphthalenecarbonitrile as L2. A further group of compounds, confirmed with

reference standards as L1, are intermediates, with various application areas. 2,3,5,6-Tetrachloropyridine and pentachloropyridine are intermediates occurring in the synthesis of the pesticides chlorpyrifos and triclopyr (Howard and Muir, 2010). Terephthalonitrile is identified as intermediate for the production of the pesticide dacthal (Meng, 2012). 2',3',4'-Trichloroacetophenone is an intermediate for the production of various fungicides and pharmaceuticals (WOC, 2019). Not much is known about the use of 2,4,6-tribromoaniline, but it might be used in the synthesis of pharmaceuticals, agricultural

pesticides and fire-extinguish agents (Labmonk, 2019). 2-Nitroanisole can have its origin in combustion processes or can be formed by atmospheric reactions (Stiborova, 2002). In 1993, large quantities of 2-nitroanisole were emitted into air during an accident at the Hoechst plant in Germany (Weyer et al., 2014). A pentachloro-methylbenzene related compound, likely a positional isomer, were detected and assigned L2, but industrial uses are not known.

### 3.5.2 Potential CEACs without LRATP

Besides those new potential CEACs with LRATP described in the previous section, we could also identify 44 new potential CEACs which do not have a predicted LRATP, according to the Stockholm Convention criteria (UNEP, 2009b), reflecting default standardised environmental conditions. Of these 44 new potential CEACs, 19 compounds were detected in the GFF sample (six as L1 and 13 as L2) and 25 compounds were detected in the PUF sample (11 as L1 and 14 as L2). An overview of L1 compounds without a predicted LRATP reflecting default environmental conditions can be found in Table 5. None of the

new L1 potential CEACs have to our knowledge been detected previously in Arctic samples, only triallate was found once before in passive air samples from Arviat, Nunavut, Canada (western shore of Hudson Bay, 61° N) (Messing et al., 2014), which is outside the Arctic circle. Triallate is an agriculture pesticide and was detected in both GFF and PUF in our sample. Four of the six L1 compounds detected in the GFF sample was also found in the PUF sample, at various GFF/PUF peak area ratios: m-Terphenyl 1:30 (GFF:PUF ratio), Triallate 1:17 (GFF:PUF ratio), Dichlofluanid 1:3 (GFF:PUF ratio) and Carbazole

1:1 (GFF:PUF ratio). The two remaining compounds, identified as L1 in the GFF sample, were 1,2-benzoanthraquinone and 6H-benzo[cd]pyren-6-one. Both are potential combustion products and can have their origin in wood or coal combustion (Czech et al., 2018; Lui et al., 2017; Vicente et al., 2016) or can be formed by atmospheric reactions (Alam et al., 2014). As





L2, we could, besides others, classify several positional isomers of reference standards which were analysed (see SI Excel-SI file for further details).

In the PUF it was possible to identify all three isomers of terphenyl (*o*, *m*, *p*) usually applied as technical mixture, while only *m*-terphenyl was detected also in the GFF. The commercial mixture of terphenyls is used as industrial agent for heat storage

and transfer as well as textile dye carriers and as intermediate of non-spreading lubricants (Netherlands, 2002). During pyrolysis and combustion of used black shorts (polyether fabric), all three terphenyl isomers were detected (Moltó et al., 2006). 4-Chloro-2-methylphenole (PCOC) is used by the industry as an intermediate for the production of phenoxy herbicides and is found as impurity in the final commercial product (B.G. Hansen et al., 2002). For dichlofluanid, carbazole, 3-iodo-2-propynyl-butylcarbamate (IPBC), and 2-(methylmercapto)benzothiazole local contamination sources cannot be excluded.

Diclofluanid and IPBC are both used as wood preservatives and and carbazole is a constituent of coal tar (creosote). In addition to that, IPBC is used in cosmetics and personal care products (ECHA, 2019f, a) and carbazole is used in the production of carbazole containing polymers (PVK, poly(-N-vinylcarbazole)) used in photovoltaic devices and in semiconducting polymers (Zhao et al., 2017; Grazulevicius et al., 2003) and pharmaceuticals (Zawadzka et al., 2015). 2-(Methylmercapto)benzothiazole is a major methylation product of 2-mercaptobenzothiazole, a common used vulcanisation accelerator in rubber of car tires,

shoes, cables, rubber gloves and toys (Herrero et al., 2014; Leng and Gries, 2017). Due to the widespread use of rubber products, in and around the sampling station, a potential local origin cannot be excluded. Dichlofluanid and carbazole was detected in both GFF and PUF sample, while IPBC and 2-(methylmercapto)benzothiazole only in the PUF sample. The mixed halogenated compound MHC-1 is an HNP emitted from marine natural sources. As earlier confirmed, the seaweed *Plocamium cartilagineum* is producing large amounts of MHC-1 (Vetter et al., 2008). MHC-1 was, however, not detected in Zeppelin air

samples reported in an earlier study (Vetter et al., 2002). Further studies are necessary to identify the origin of MHC-1 in the Arctic. No information was found on the industrial usage of 2-bromo-3,5-dimethoxytoluene, but formation as HNP cannot be excluded, since chlorinated dimethoxytoluenes were previously identified in lichen (Elix et al., 1984).

### 3.5.3 Estimated half-life's in air reflecting Arctic environmental conditions

Our $t_{1/2}$(air) is based on default values retrieved from EPIsuite (U.S.EPA, 2019). Standardised estimates are commonly used

for the estimation of LRATP (Muir and Howard, 2006; Howard and Muir, 2010; Brown and Wania, 2008; Reppas-Chrysovitsinos et al., 2017). These default half-lives are likely underestimated when adjusted to Arctic environmental conditions. When adjusting the estimates of $t_{1/2}$(air) for the sampling temperature and assumed OH radical concentrations in December (see sect. 2.5), all compounds, classified as L1 and L2 have an estimated $t_{1/2}$(air), exceeding 2 days. Results for selected compounds can be found in Table 6 and further results in SI Table S3 and Excel-SI. This supports our assumption

that those new potential CEACs could be subject to LRAT as a result of enhanced persistence in air during Arctic winter. While influences from nearby sources cannot be excluded, those properties are relevant for 2 out of 4 hazard criteria defining a POP, according to the Stockholm convention (UNEP, 2009b), suggesting they deserve further focus from the research and policy communities. While the selected numerical values used to predict adjusted reaction half-lives may be questioned, these

data in combination with their findings in Arctic air samples suggest that LRATP cannot be excluded. While half-lives are prolonged under relevant Arctic conditions, we caution that our estimates do not account for differences in net atmospheric deposition among the substances studied which may limit LRATP (e.g. (Beyer et al., 2003)).

### 3.5.4 Comparison of findings in Arctic air to air samples from southern Norway

For some compounds it was possible to compare findings from this study of Arctic air samples to findings of similar high-volume air samples from Birkenes in southern Norway (Röhler et al., 2020). The Birkenes observatory is a part of EMEPs monitoring stations for background air, and the air samples were collected during April–May 2015. For a complete overview of compounds that were identified both studies, see Excel file SI. Among the new potential CEACs detected in Arctic air, it was possible to find five of 15 L1 compounds with LRAT and 10 of 13 L1 compounds without LRAT also in the Birkenes air.

The identification of new potential CEACs in air samples from both southern Norway (Birkenes) and the Arctic (Zeppelin, Svalbard), combined with predictions of $t_{1/2}$(air) which are adjusted to reflect actual environmental conditions, supports our assumption that these compounds may undergo LRAT.

### 3.6 Summary for Level 3 compounds

A large number of L3 compounds, tentative candidates, were detected in the Arctic air samples. The bulk of them are PACs,
primarily PAHs, substituted PAHs (e.g. alkane side chains), halogenated PAHs and sulphur- nitrogen- and oxygen-containing PAHs (Figure 6). The tentatively identified compounds also include several phthalates, carbonic acid esters, and miscellaneous halogenated compounds. The list of L3 compounds can be found in SI (Excel-SI).

### 3.7 Level 4 compounds

The group of L4 compounds includes compounds with an assigned molecular formula and several unknown halogenated
compounds, which did not match any of the MS in the used MS libraries. The approximate molecular weight (nominal mass), the degree of halogenation, and some major fragments could be extracted from the LRMS spectra (see SI Excel-SI). Additional structural information was obtained using GC×GC-HRMS for some of the unknown halogenated compounds.

The acquired accurate mass spectra from HRMS (see SI for HRMS spectra) were processed using MetFrag software (MetFrag, 2019; Ruttkies et al., 2016) and possible molecular formula/s were generated (Table 7).
After searching SciFinder® with possible molecular formulas and identified substructures from the mass spectra, it was possible to find possible structures suggestions for several of the unknown halogenated compounds analysed with HRMS. The number of citations of a compound in SciFinder could give a further limitation of possible structures. Since the mass spectra do not occur in the NIST14 MS library, the found compound might be a less cited compound or might not have registered/ assigned with a CAS number and is not yet listed in the CAS registry in SciFinder. Using HRMS and SciFinder data, additional structural
information could be extracted for four unknown halogenated compounds (Table 7 and SI Fig. S2-S7), originally classified as L4. Two of the compounds were tentatively identified as methoxylated halogenated benzenes, one dibromo-monochloro-



anisole and one dichloro-methyl-dimethoxy-benzene. Several structurally related compounds were found among the potential CEACs with a default LRATP (see sect. 3.5.1 and Table 4) of which one, chloroneb, was assigned L1 confidence, which supports the tentative structure assignments and qualify the two for L3.

## 4. Conclusions

By applying a dedicated non-target and suspect screening method based on a non-destructive sample clean-up method (excluding acid treatment) combined with GC×GC-LRMS on high-volume air samples from Arctic Svalbard, a large number of known and new potential CEACs could be identified at prioritised. During this study, 73 new potential CEACs (compounds previously not reported in Arctic environments) were classified at confidence level L1 or L2, which indicate that comprehensive suspect and non-target screening can reveal new potential CEACs that might be needed to be monitored or risk assessed. All these compounds are predicted to have atmospheric reaction half-lives exceeding two days, if these are adjusted to reflect actual environmental conditions during sampling. Reaction half-lives reflecting standardised environmental conditions (e.g. 25 °C) are, thus, poor predictors for persistence in the Arctic environment. The here reported study underpins the importance of combining model estimates with empirical measurements for environmental assessment of chemicals. The newly identified organic CEACs from this study are recommended for inclusion in regulatory monitoring strategies and for target specific analytical methods. Although the applied identification method is a promising tool for identification of new priority pollutants, but we do not consider the current study as exhaustive. Further in-depth studies, carried out using GC×GC-HRMS are expected to provide additional information about CEACs not yet included in MS libraries. Those should preferably use a column set featuring a non-polar first dimension column, which allow comparisons to retention time databases or retention index prediction data (Veenaas and Haglund, 2018) in order to accept or reject the candidate structures of hitherto unknown CEACs.

**Competing interests**

The authors declare that they have no conflict of interest.

**Acknowledgment**

Compound structures were created using ChemOffice19 (PerkinElmerInformatics, 2019).
LogP and logD values were created using JChem for Excel (ChemAxon, 2019).



**Author contribution**

LR, MS, PBN and RK developed the idea behind this study.

LR performed chemical work, analysis, created the figures and wrote the paper.

MS and PBN provided guidance and contributed to the paper preparation

PH provide guidance, HRMS measurements and contributed to the paper preparation

KB provided guidance on theoretical calculations and contributed to the paper preparation

RK provided financial support, academic guidance and contributed to the paper preparation

All authors read and approved the submitted manuscript.

**Financial support**

This study was funded by NMBU, Norwegian University of Life Sciences, Ås with an internal PhD grant, NILU, Norwegian
Institute for Air Research, Kjeller and the Norwegian Ministry of Climate and Environment through two Strategic Institute
Programs, granted by the Norwegian Research Council ("Speciation and quantification of emerging pollutants" and "New
measurement methods for emerging organic pollutants"). KB received support from the Research Council of Norway
(#267574).

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





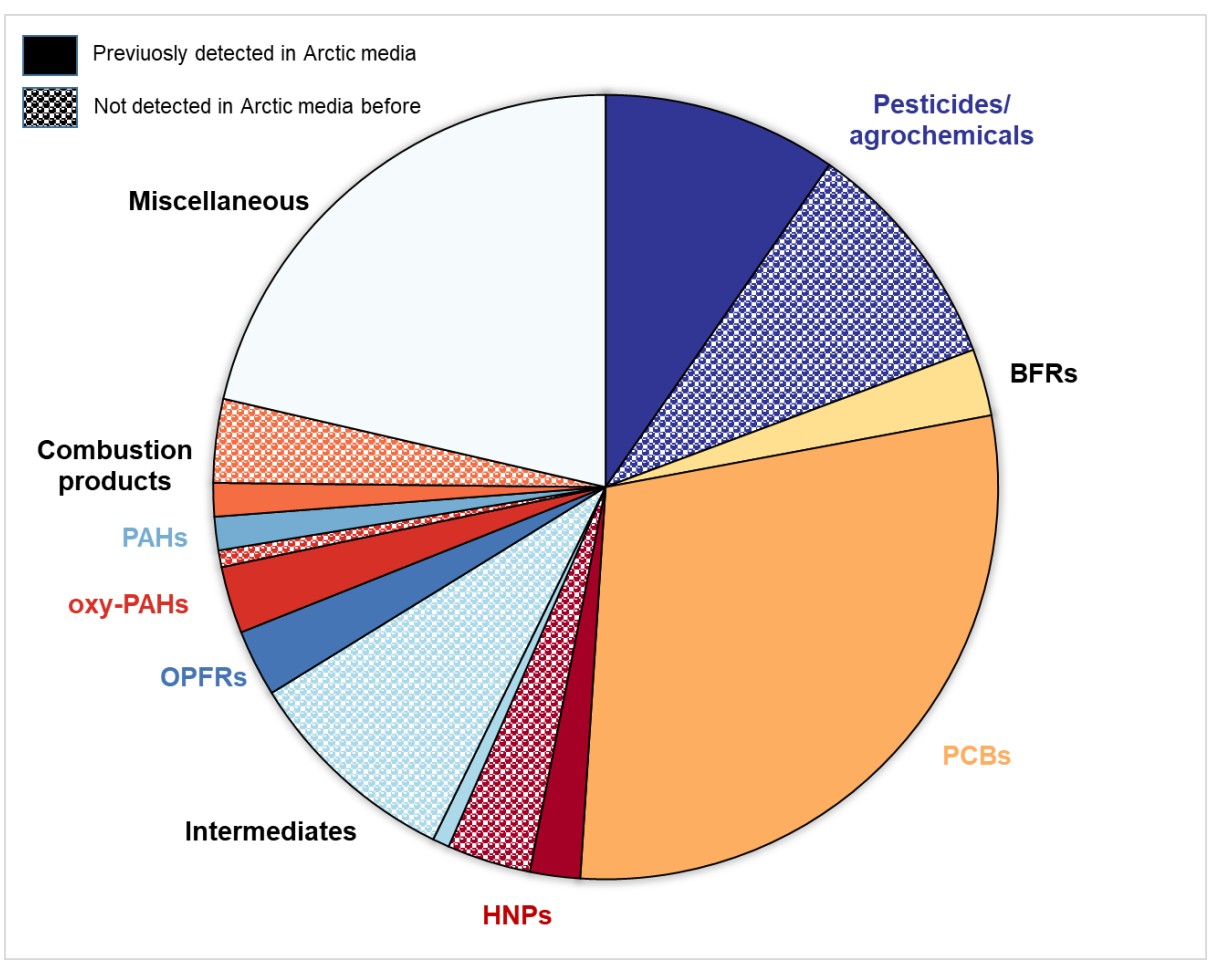

Figure 1: Graphical abstract, summary of compounds confirmed with reference standards and compounds with tentative structure.





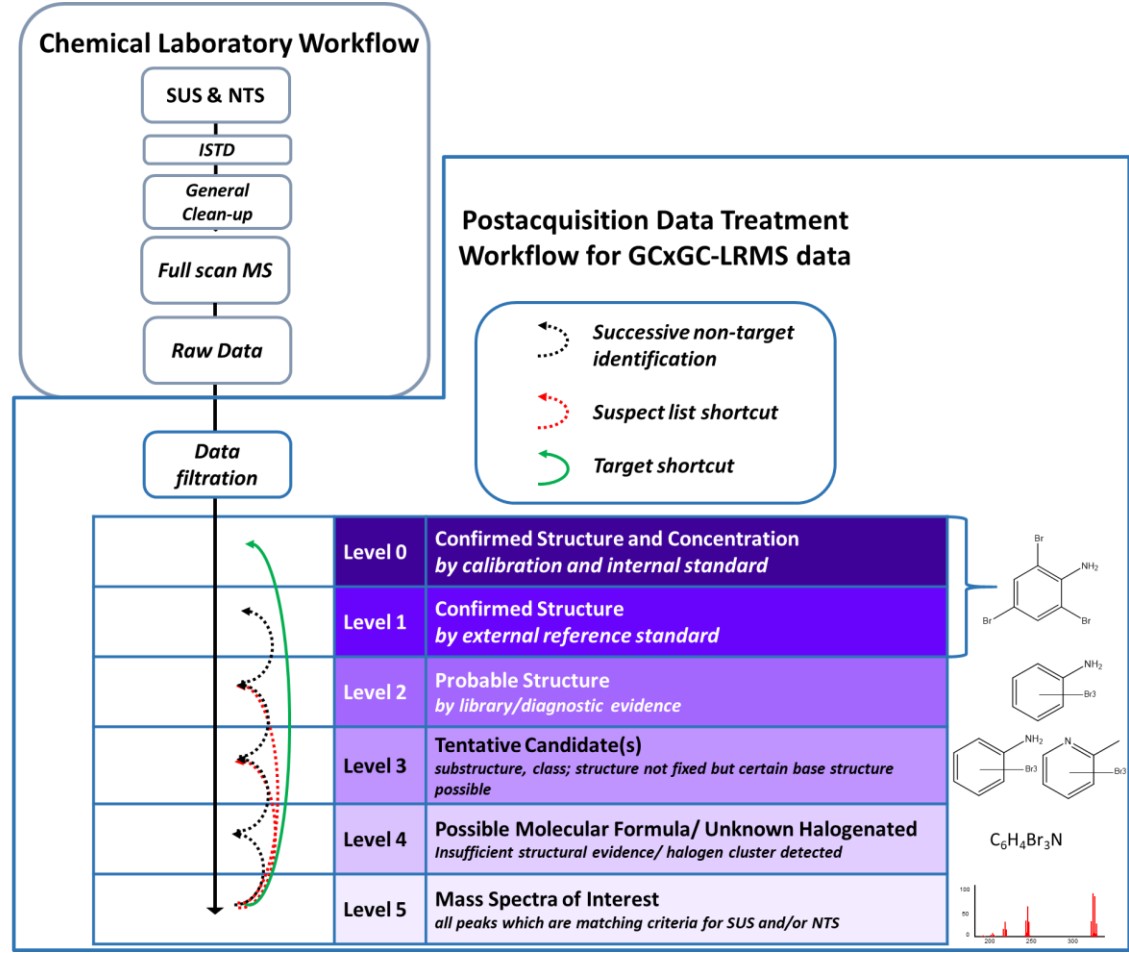

**Figure 2: General strategy and identification confidence for GC×GC-LRMS. Adapted from Schymanski et al. (2015) and Röhler et al. (2020).**





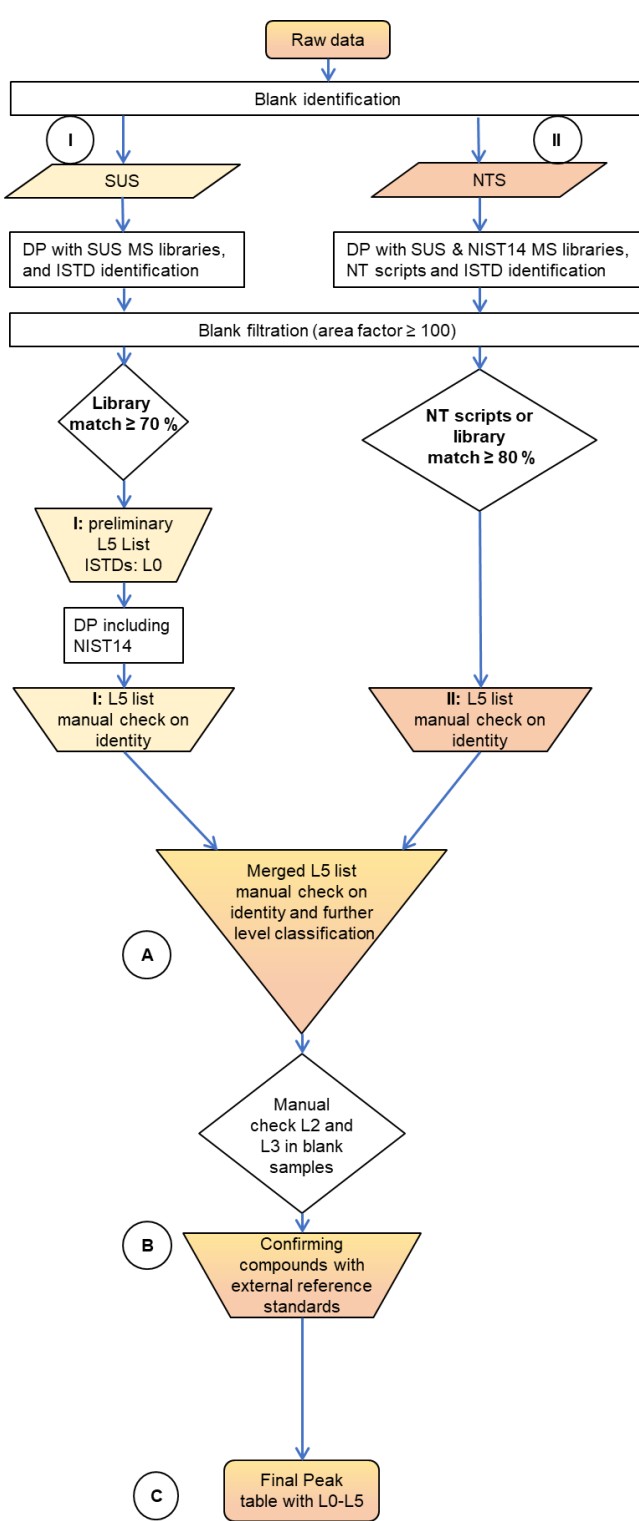

**Figure 3: Data Processing workflow for suspect and non-target screening.**




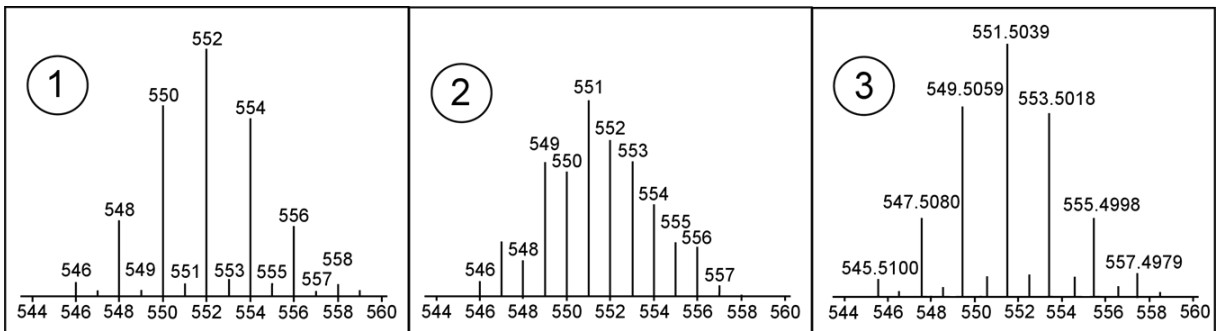

**Figure 4: 1: Isotope cluster of hexabromobenzene (HBB) in NIST14, 2: own measured HBB on GC×GC-LRMS and 3: HRMS isotope cluster HBB (Röhler et al., 2020)**





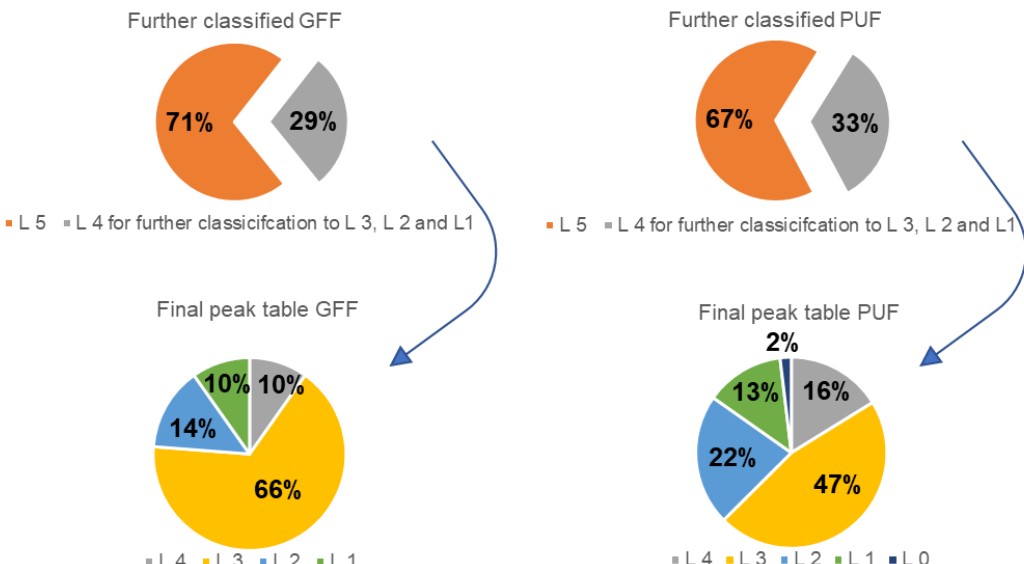

**Figure 5: Distribution of L0-L5 compounds in the GFF and PUF sample.**





**Table 1: Overview of the L0–L4 compounds, classified in Arctic air samples.**

| Level | Compounds classified | PUF sample | GFF sample | Common to PUF and GFF | Found in suspect lists |
|---|---|---|---|---|---|
| **L0** | 6 | 6 | Only traces detected | 0 | 1 |
| **L1** | 56 | 42 | 14 | 7 | 39 |
| **L2** | 90 | 70 (41 PCBs) | 20 | 0 | 17[a] |
| **L3** | 241 | 146 | 95 | 0 | -[b] |
| **L4** | 65 | 51 (29 unknown halogenated) | 14 (12 unknown halogenated) | 0 | -[b] |

[a] showing similarity to suspect lists, isomer not confirmed; [b] not applicable



**Table 2: : Ranking of most abundant POPs in this study (based on peak area) in comparison to concentrations from target analysis (pg m-3) in the Norwegian national monitoring programme of long-range transported environmental contaminants (Nizzetto, 2016).**

| Compound | Area from this study | Average concentration in December 2015 at Zeppelin [pg m$^{-3}$](Nizzetto, 2016) |
|---|---|---|
| HCB | 8032400 | 80.8 |
| PeCB | 890100 | 25.1[a] |
| α-HCH | 652200 | 3.25 |
| *p,p'*-DDE | 297500 | 0.89 |
| γ-HCH | 177700 | 0.6 |
| *o,p'*-DDT | 46700 | 0.16 |
| Dieldrin | 37700 | _[b] |
| *trans*-Nonachlor | 36900 | 0.37 |
| *cis*-Chlordane | 36100 | 0.35 |
| Heptachloro *exo* epoxide | 25800 | _[b] |
| *p,p'*-DDT | 18800 | 0.11 |
| PCB-153 | 15100 | 0.15 |
| PBDE-47 | 9800 | 0.07 |
| PBDE-28 | 600 | 0.006 |

*[a]: Not shown in report; [b]: Non-acid stable compound and not included in the Norwegian national air monitoring*





**Table 3: Structure overview of L1 compounds, classified as new potential CEACs with LRATP**

| Name/ CAS/ Sample | Structure | Name/ CAS/ Sample | Structure |
|---|---|---|---|
| Benzenesulfonamide (BSA)/ 98-10-2 GFF (particle phase) | | 2-Naphthalenecarbonitrile/ 613-46-7 PUF (gas phase) | |
| 2-Methyl-9,10-Anthraquinone/ 84-54-8 GFF (particle phase) | | 2,3,5,6-Tetrachloropyridine/ 2402-79-1 PUF (gas phase) | |
| 2,6-Dichlorobenzonitrile (dichlorobenil)/ 1194-65-6 PUF (gas phase) | | Pentachloropyridine/ 2176-62-7 PUF (gas phase) | |
| 2,4-Dichlorobenzonitrile/ 6574-98-7 PUF (gas phase) | | 1,4-Benzenedicarbonitrile (Terephthalonitrile)/ 623-26-7 PUF (gas phase) | |
| 1,4-Dichloro-2,5-dimethoxybenzene (chloroneb)/ 2675-77-6 PUF (gas phase) | | 2',3',4'-Trichloroacetophenone / 13608-87-2 PUF (gas phase) | |
| 2-Chloro-6-(trichloromethyl)pyridine (Nitrapyrin)/ 1929-82-4 PUF (gas phase) | | 2,4,6-Tribromoaniline / 147-82-0 PUF (gas phase) | |
| 2,4-Dichloroanisole/ 553-82-2 PUF (gas phase) | | 2-Nitroanisole/ 91-23-6 PUF (gas phase) | |
| 2,4,6-Trichloroanisole/ 87-40-1 PUF (gas phase) | | | |



**Table 4: Overview of L2 compounds, classified as new potential CEACs with LRATP.**

| Name | Sample | Molecular formula |
|---|---|---|
| 3,4-Dichloropropiophenone related positional isomer[a] | GFF (particle phase) | $C_9H_8Cl_2O$ |
| Diphenyl sulfone | GFF (particle phase) | $C_{12}H_{10}O_2S$ |
| Dibenzothiophene sulfone | GFF (particle phase) | $C_{12}H_8O_2S$ |
| N-(2-Cyanoethyl)-N-methyl-benzenesulfonamide | GFF (particle phase) | $C_{10}H_{12}N_2O_2S$ |
| Two chloroneb related positional isomers[b] | PUF (gas phase) | $C_8H_8Cl_2O_2$ |
| One chlorothalonil related positional isomer[c] | PUF (gas phase) | $C_8Cl_4N_2$ |
| Two trichloro-dimethoxybenzen isomers | PUF (gas phase) | $C_8H_7Cl_3O_2$ |
| Two dichloro-methylanisole isomers | PUF (gas phase) | $C_8H_8Cl_2O$ |
| One dibromo-dimethoxybenzene isomer | PUF (gas phase) | $C_8H_8Br_2O_2$ |
| 1-naphthalenecarbonitrile | PUF (gas phase) | $C_{11}H_7N$ |
| One pentachloro-methylbenzene positional isomer[d] | PUF (gas phase) | $C_7H_3Cl_5$ |

[a] Retention times close to, but not identical to, that of a 3,4-dichloropropiophenone standard

[b] Retention times close to, but not identical to, that of a chloroneb standard

[c] Retention times close to, but not identical to, that of a chlorothalonil standard

5  [d] Retention times close to, but not identical to, that of a pentachlorotoluene standard



**Table 5: Structure overview of L1 compounds, classified as new potential CEACs without a predicted LRATP under standardised environmental conditions.**

| Name/ CAS/ Sample | Structure | Name/ CAS/ Sample | Structure |
|---|---|---|---|
| 1,2-Benzanthraquinone/ 2498-66-0 GFF (particle phase) | | *p*-Terphenyl/ l92-94-4 PUF (gas phase) | |
| 6H-Benzo[*cd*]pyren-6-one/ 3074-00-8 GFF (particle phase) | | 4-Chloro-2-methylphenole (PCOC)/ 1570-64-5 PUF (gas phase) | |
| Triallate/ 2303-17-5 GFF and PUF | | 3-Iodo-2-propynyl-butylcarbamate (Iodocarb, IPBC)/ 55406-53-6 PUF (gas phase) | |
| Dichlofluanid/ 1085-98-9 GFF and PUF | | 2-(Methylmercapto)-benzothiazole/ 615-22-5 PUF (gas phase) | |
| Carbazole/ 86-74-8 GFF and PUF | | MHC-1 (2-bromo-1-bromomethyl-1,4-dichloro-5-(2′-chloroethenyl)-5-methylcyclohexane)/ 66321-24-2 PUF (gas phase) | |
| *m*-Terphenyl/ l92-06-8 GFF and PUF | | 2-bromo-3,5-dimethoxytoluene/ 13321-73-8 PUF (gas phase) | |
| *o*-Terphenyl/ 84-15-1 PUF (gas phase) | | | |





**Table 6: Half-life in air: Standard values from EPIsuite and adjusted for Arctic conditions (Eq.S1-S2), for selected compounds.**

| Name | CAS | Standard half-life [days] (25 °C; 1.5E6 mol cm$^{-3}$) | Adjusted half-life [days] (-2.4 °C; 6.0E3 mol cm$^{-3}$) |
|---|---|---|---|
| 9-Fluorenone | 486-25-9 | 1.7 | 651 |
| *p,p'*-DDE | 72-55-9 | 1.4 | 541 |
| Dieldrin | 60-57-1 | 1.2 | 437 |
| 1,9-Benz-10-anthrone | 82-05-3 | 0.6 | 223 |
| Caffeine | 58-08-2 | 0.6 | 207 |
| TCIPP | 13674-84-5 | 0.2 | 90 |
| TCEP | 115-96-8 | 0.5 | 183 |
| Benzo[*ghi*]fluoranthene | 203-12-3 | 0.2 | 65 |
| Naphthalene | 91-20-3 | 0.5 | 186 |
| Tris(3-chloropropyl) phosphate | 1067-98-7 | 0.1 | 55 |
| *m*-Terphenyl | 92-06-8 | 0.8 | 159 |
| Dichlofluanid | 1085-98-9 | 0.7 | 135 |
| IPBC | 55406-53-6 | 0.4 | 79 |





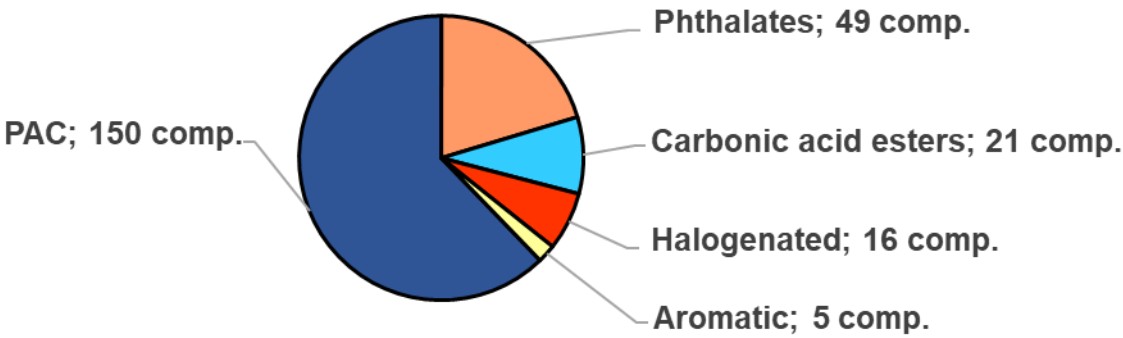

**Figure 6: L3 compound groups.**





**Table 7: Unknown halogenated compounds with HRMS data.**

| Compound | Accurate mass | Possible molecular formula from MetFrag | Formula supported by manual fragment interpretation |
|---|---|---|---|
| A#9842 GFF | 256.0169 | $C_{11}H_{10}Cl_2N_2O$ | $C_{11}H_{10}Cl_2N_2O$ |
| B#11108 GFF | 230.0134 | $C_8H_8Cl_2N_4$ | m/z 230, dichloro- fragment $C_{10}H_{10}Cl_2NO$ |
| C#4444 PUF | 299.8372 | $C_7H_5Br_2ClO$ $C_6H_5Br_2O_2P$ | $C_7H_5Br_2ClO$ |
| D#5672 PUF | 220.0053 | $C_9H_{10}Cl_2O_2$ $C_8H_{10}ClO_3P$ | $C_9H_{10}Cl_2O_2$ |