# Peer review of "Non-target and suspect characterisation of organic contaminants in Arctic air, Part II"

_Atmospheric Chemistry and Physics, 2020_

## Referee Comment (RC1) · Anonymous Referee #1 · 25 Feb 2020

I liked the paper. It is done on the high level both in terms of finding classic pollutants and identification of chemicals of emerging concern in Arctic air. The study looks pretty solid addressing relevant scientific questions dealing with atmospheric chemistry. A lot of new interesting data allows the authors proposing valuable conclusions and ideas for future studies. The authors cite a number of publications reviling earlier results. The title nicely reflects the essence of the study while the abstract provides the crucial information on the work completed. The manuscript is easy to read. No problems with

language. Besides, the text is well illustrated. Surely it would be better to use high resolution instrument. Nevertheless the authors tried to extract the maximum information from the low resolution mass spectra. I found many novel interesting structures, which may be quite helpful in future studies. I did not find serious mistakes requiring major revision of the manuscript. Below are just two comments. Page 5, sect.30 and further - SUS and NTS data processing reduced the number of peaks requiring manual interpretation. How many peaks passed that stage? Were all the peaks which did not pass that process checked manually? The authors mention as difficult cases only poly-halogenated compounds, however due to coelution and low levels of many constituents their spectra are quite often far from being ideal. Manual interpretation is always useful. Page 6, sect.5 and conclusions, sect.20 - Mass spectrometrists often forget about the usefulness of the retention indices information. Nevertheless in many cases it may help a lot, providing crucial information or making structural elucidation more reliable. Unfortunately that approach was not used in the described study. The authors mentioned that the first column RT were not applied due to the specific type of the first column. In conclusions it is emphasized that in the further in-depth GCxGC-HRMS study a non-polar first dimension column, allowing application of the RT databases and RI prediction data, will be used. I would like to mention that second-dimension retention indices may be quite useful as well when using comprehensive two-dimensional gas chromatography (Mazur et al, J.Chromatogr. A, 2018, 1569, 178-185).

---

## Referee Comment (RC2) · Anonymous Referee #2 · 12 Mar 2020

This study identifies and prioritises known and potential new chemicals of emerging Arctic concern in two pooled high-volume air samples from the Arctic background monitoring station Zeppelin (Svalbard). In total, 73 compounds previously not reported in Arctic environments were classified at high confidence level using a non-destructive wide scope clean-up method combined with GCxGC-LRMS for a suspect and non-target screening approach. The focus of this paper is the application of the method to Arctic samples and the data processing, whereas the authors refer to their companion

paper for details on the development of the analytical method ("Non-target and suspect characterisation of organic contaminants in ambient air, Part I: Combining a novel sample clean-up method with comprehensive two-dimensional gas chromatography").

The large number of identified or tentatively identified compounds that have not been reported in Arctic environments before underlines the significance of this study and the importance of comprehensive suspect and non-target screening approaches. The findings can be incorporated in discussions for future monitoring programs and the development of targeted analytical methods. As the majority of suspect and non-target screening studies are based on LC-HRMS approaches, the chosen GCxGC-LRMS technique in combination with the wide scope clean-up offers a new perspective and a focus on different compounds. In addition, it is of specific interest that the study deals with samples from a remote region.

As the manuscript is scientifically sound, well structured and well written, I only have some minor comments/suggestions. - Figure 1: You list the numbers in the text, but it would be easier to grasp if you add the number of compounds included in the pie chart (and its different sections) in the graphical abstract as well. - Page 2, lines 2/3: There is a new paper from Wang et al. listing even more compounds (Environ. Sci. Technol. 2020, 54, 2575−2584, https://dx.doi.org/10.1021/acs.est.9b06379). Maybe it's worth to include it as reference? - Page 5, line 29/Figure 3: Based on which criteria does the software calculate the forward match percentage? This could be an important information to include as it is a major filtering step. - Page 6, lines 17/18: You say that an area threshold of 100 was chosen as areas are not adjusted for sample volumes. Could you mention how the volumes of sample and blanks differed? - Page 8, line 23/Table 2: If I get it correctly, at least two compounds in table 2 were classified as level 0 for which target quantification could be done. Are the determined concentrations in a similar range compared to the average data from Nizzetto et al.?

Technical corrections: The text is written very well, but still contains some typing/auto correction errors. Things I noticed while reading: - Page 1, line 13: possesses? - Page

1, line 14: "sparsely populated" doesn't fit grammatically - Figure 1: previously - Page 2, line 21: Air Pollution - Page 4, line 4: "in the" 2x - Page 5, line 17: word after slightly is missing - Page 8, lines 4 to 7: nested sentence, difficult to understand - Page 8, line 18: peak area - Page 8, line 29: that also has - Page 9, line 1: an/one isomer of TCEP? - Page 10, line1: emerging Arctic concern - Page 12, line 23: half-lives - Page 14: could be identified and prioritised? - Page 23, caption figure 3: Data processing

---

## Author Comment (AC1) · 7 May 2020

Kjeller, May 7th, 2020

ISSUE: Reply to reviewer comments on Manuscript ACP acp-2020-105 entitled "Non-target and suspect characterisation of organic contaminants in Arctic air, Part II: Application of a new tool for identification and prioritisation of chemicals of emerging Arctic concern in air

[Figure]

Dear Editor, Thank you and the anonymous reviewers a for the constructive and helpful comments and suggestions on our manuscript. Please find enclosed our detailed reply letter on the reviewer comments to our manuscript "Non-target and suspect characterisation of organic contaminants in Arctic air, Part II: Application of a new tool for identification and prioritisation of chemicals of emerging Arctic concern in air". All suggested changes listed in the reviewer replies are comprehensively addressed. After discussion within the author team, we have listed our final recommendations and suggestions below.

We wish to thank the reviewers for insightful and constructive comments and hope that our response is in accordance with their expectation.

Sincerely yours

Laura Röhler On behalf of the author team

Reply to reviewer comments Anonymous Referee #1 Reviewer comments: I liked the paper. It is done on the high level both in terms of finding classic pollutants and identification of chemicals of emerging concern in Arctic air. The study looks pretty solid addressing relevant scientific questions dealing with atmospheric chemistry. A lot of new interesting data allows the authors proposing valuable conclusions and ideas for future studies. The authors cite a number of publications reviling earlier results. The title nicely reflects the essence of the study while the abstract provides the crucial information on the work completed. The manuscript is easy to read. No problems with language. Besides, the text is well illustrated. Surely it would be better to use high resolution instrument. Nevertheless, the authors tried to extract the maximum information from the low resolution mass spectra. I found many novel interesting structures, which may be quite helpful in future studies. I did not find serious mistakes requiring major revision of the manuscript. Below are just two comments.

Page 5, sect.30 and further - SUS and NTS data processing reduced the number of peaks requiring manual interpretation. How many peaks passed that stage? Were all

the peaks which did not pass that process checked manually? The authors mention as difficult cases only polyhalogenated compounds, however due to coelution and low levels of many constituents their spectra are quite often far from being ideal. Manual interpretation is always useful.

- How many peaks passed that stage?

Author reply The raw data set of the GFF contained over 16 000 features and almost 20 000 features for the PUF. These numbers are not presented in the paper, but could be calculated from the information given at page 7 sect.10. There we discuss the outcome of the data processing, we state that over 700 compounds were classified as L5 in GFF, and over 1200 compounds in PUF. In addition, we refer to Fig. S1 in the SI for peak reduction during data processing.

We included the relevant numbers at page 7 sect. 12: ...out of over 16000 features in GFF and almost 20000 features for PUF respectively...

Since page 5, sect.30 is dealing with background information on the data processing we did not consider the inclusion of an extended number of signals as feasible and helpful for the reader. However, we included at page 5, sect. 30 a reference to section 3.1.: ...(see sect. 3.1. for numbers)...

Reviewer comment - Were all the peaks which did not pass that process checked manually? Author reply: Manual inspection of almost 40000 features on right identity for level classification was considered as not adequate and excessively time consuming. Hence, only peaks which passed these stages were manually inspected. This list contained over 700 compounds in GFF and over 1200 in PUF, as is stated in the paper. This procedure was considered the best compromise between not missing important compounds and too many features for manual check on right identity.

Reviewer comment Page 6, sect.5 and conclusions, sect.20 - Mass spectrometrists often forget about the usefulness of the retention indices information. Nevertheless,

in many cases it may help a lot, providing crucial information or making structural elucidation more reliable. Unfortunately, that approach was not used in the described study. The authors mentioned that the first column RT were not applied due to the specific type of the first column. In conclusions it is emphasised that in the further in-depth GCxGC-HRMS study a non-polar first dimension column, allowing application of the RT databases and RI prediction data, will be used. I would like to mention that second-dimension retention indices may be quite useful as well when using comprehensive two-dimensional gas chromatography (Mazur et al, J.Chromatogr. A, 2018, 1569, 178-185).

Author reply Thank you for the interesting reference. The author team appreciates this information. We acknowledge that both indices for column 1 and 2 (for first and second-dimension retention), are of great value during GC×GC separation. Unfortunately, the application of retention indices was not possible on this data set since the development of an adapted/new RI system in order to fit our column combination and not only for the first dimension. However, we included Mazur et al. as reference for retention indices for GC×GC at page 6 sect.8.

Anonymous Referee #2 Reviewer comments: This study identifies and prioritises known and potential new chemicals of emerging Arctic concern in two pooled high-volume air samples from the Arctic background monitoring station Zeppelin (Svalbard). In total, 73 compounds previously not reported in Arctic environments were classified at high confidence level using a non-destructive wide scope clean-up method combined with GCxGC-LRMS for a suspect and nontarget screening approach. The focus of this paper is the application of the method to Arctic samples and the data processing, whereas the authors refer to their companion paper for details on the development of the analytical method ("Non-target and suspect characterisation of organic contaminants in ambient air, Part I: Combining a novel sample clean-up method with comprehensive two-dimensional gas chromatography"). The large number of identified or tentatively identified compounds that have not been reported in Arctic environments

before underlines the significance of this study and the importance of comprehensive suspect and non-target screening approaches. The findings can be incorporated in discussions for future monitoring programs and the development of targeted analytical methods. As the majority of suspect and non-target screening studies are based on LC-HRMS approaches, the chosen GCxGC-LRMS technique in combination with the wide scope clean-up offers a new perspective and a focus on different compounds. In addition, it is of specific interest that the study deals with samples from a remote region. As the manuscript is scientifically sound, well structured and well written, I only have some minor comments/suggestions.

- Figure 1: You list the numbers in the text, but it would be easier to grasp if you add the number of compounds included in the pie chart (and its different sections) in the graphical abstract as well. Author reply : All compound numbers for each section have been included in Figure 1.

Reviewer comment - Page 2, lines 2/3: There is a new paper from Wang et al. listing even more compounds (Environ. Sci. Technol. 2020, 54, 2575ôĂĂĂ2584, https://dx.doi.org/10.1021/acs.est.9b06379). Maybe it's worth to include it as reference? Author reply: Thank you for the reference. Since we wanted to illustrate how fast the number of registered chemicals in CAS is rising as comprehensively recorded under https://www.cas.org/support/documentation/chemical-substances, we still consider the CAS 100 million registry benchmark compared to today's numbers as an important illustration of the importance for chemicals in our western societies.

Reviewer comment: - Page 5, line 29/Figure 3: Based on which criteria does the software calculate the forward match percentage? This could be an important information to include as it is a major filtering step. Author reply: ChromaToF is using the composite algorithm of NISTs MSsearch engine (Samokhin et al. https://doi.org/10.1002/jms.3591), and this information is now included in the manuscript. "the forward match percentage to the mass spectrum (MS) library (ChromaToF is using NISTs composite algorithm, c.f. (Samokhin et al., 2015)). . ."

[Figure]

Reviewer comments: - Page 6, lines 17/18: You say that an area threshold of 100 was chosen as areas are not adjusted for sample volumes. Could you mention how the volumes of sample and blanks differed? Author reply: Sample volumes are visually adjusted, which means the extract volume was visually adjusted to the same height in the final vial. Here, differences could have happened since these glass vials are not volumetrically calibrated. We included this information in the text: " The different sample extracts were visually adjusted to the same height, before taking out aliquots for GC×GC analysis (uncertainty ± 10 %)"

Reviewer comments: - Page 8, line 23/Table 2: If I get it correctly, at least two compounds in table 2 were classified as level 0 for which target quantification could be done. Are the determined concentrations in a similar range compared to the average data from Nizzetto et al.? Author reply This assumption is correct, HCB and p,p'-DDT are L0 compounds since they are part of our ISTD mixture for quality control. Since we were only using the ISTDs for quality assurance, and not for quantification, we did not include quantification standards in our sample analysis, but are planning to do that on future projects. So unfortunately, we cannot determine concentrations for these compounds in the here presented study.

Reviewer comments: Technical corrections: The text is written very well, but still contains some typing/auto correction errors. Things I noticed while reading: - Page 1, line 13: possesses? Author reply: Corrected

Reviewer comment: – Page 1, line 14: "sparsely populated" doesn't fit grammatically Author reply: Corrected to: sparse population

Reviewer comment: - Figure 1: previously Author reply: Corrected

Reviewer comment: – Page 2, line 21: Air Pollution Author reply: Corrected

Reviewer comment: - Page 4, line 4: "in the" 2x Author reply: Corrected

Reviewer comment: - Page 5, line 17: word after slightly is missing Author reply: Corrected: slightly adjusted

Reviewer comment: - Page 8, lines 4 to 7: nested sentence, difficult to understand Author reply: Corrected:". . .. into four groups. These groups are (i) legacy POPs and PAHs. . .."

Reviewer comment: - Page 8, line 18: peak area Author reply: Corrected

Reviewer comment: - Page 8, line 29: that also has Author reply: Corrected

Reviewer comment: - Page 9, line 1: an/one isomer of TCEP? Author reply: Corrected: one isomer of TCEP

Reviewer comment. - Page 10, line1: emerging Arctic concern Author reply: Corrected

Reviewer comment: - Page 12, line 23: half-lives Author reply: Corrected

Reviewer comment: – Page 14: could be identified and prioritised? Author reply: Corrected

Reviewer comment - Page 23, caption figure 3: Data processing Author reply: Corrected